# Global analysis of cytosine and adenine DNA modifications across the tree of life

Sreejith Jayasree Varma[1†], Enrica Calvani[2,3†], Nana-Maria Grüning[1], Christoph B Messner[2,3], Nicholas Grayson[4], Floriana Capuano[3], Michael Mülleder[5], Markus Ralser[1,2,3]*

[1]Department of Biochemistry, Charité Universitätsmedizin, Berlin, Germany; [2]The Molecular Biology of Metabolism Laboratory, The Francis Crick Institute, London, United Kingdom; [3]Department of Biochemistry and Cambridge Systems Biology Center, University of Cambridge, Cambridge, United Kingdom; [4]Wellcome Trust Sanger Institute, Wellcome Trust Genome Campus, Hinxton, United Kingdom; [5]Core Facility-High Throughput Mass Spectrometry, Charité Universitätsmedizin, Berlin, Germany

**\*For correspondence:**
markus.ralser@charite.de

[†]These authors contributed equally to this work

**Competing interest:** The authors declare that no competing interests exist.

**Abstract** Interpreting the function and metabolism of enzymatic DNA modifications requires both position-specific and global quantities. Sequencing-based techniques that deliver the former have become broadly accessible, but analytical methods for the global quantification of DNA modifications have thus far been applied mostly to individual problems. We established a mass spectrometric method for the sensitive and accurate quantification of multiple enzymatic DNA modifications. Then, we isolated DNA from 124 archean, bacterial, fungal, plant, and mammalian species, and several tissues and created a resource of global DNA modification quantities. Our dataset provides insights into the general nature of enzymatic DNA modifications, reveals unique biological cases, and provides complementary quantitative information to normalize and assess the accuracy of sequencing-based detection of DNA modifications. We report that only three of the studied DNA modifications, methylcytosine (5mdC), methyladenine (N6mdA) and hydroxymethylcytosine (5hmdC), were detected above a picomolar detection limit across species, and dominated in higher eukaryotes (5mdC), in bacteria (N6mdA), or the vertebrate central nervous systems (5hmdC). All three modifications were detected simultaneously in only one of the tested species, *Raphanus sativus*. In contrast, these modifications were either absent or detected only at trace quantities, across all yeasts and insect genomes studied. Further, we reveal interesting biological cases. For instance, in *Allium cepa*, *Helianthus annuus*, or *Andropogon gerardi*, more than 35% of cytosines were methylated. Additionally, next to the mammlian CNS, 5hmdC was also detected in plants like *Lepidium sativum* and was found on 8% of cytosines in the *Garra barreimiae* brain samples. Thus, identifying unexpected levels of DNA modifications in several wild species, our resource underscores the need to address biological diversity for studying DNA modifications.

## Editor's evaluation

DNA methylation is an important mechanism to control gene expression, yet methods for quantitation of global DNA methylation analyses are limited. This work provides a new sensitive method for the quantitation of global DNA methylation and they apply this to over 100 species of eukaryotes and prokaryotes, finding interesting differences across species. This is a useful tool and resource for those interested in DNA methylation and evolution.

## Introduction

Enzyme-catalyzed DNA modifications are studied for their roles in chromatin structure, gene-expression regulation, prevention of viral DNA integration, epigenetic inheritance, cell–environment interactions, developmental biology, immunity, memory, aging, and cancer (*Miller and Grant, 2013*; *Breiling and Lyko, 2015*; *Guo et al., 2011*; *Jessop et al., 2018*; *de la Calle-Fabregat et al., 2020*; *Day and Sweatt, 2010*; *Masser et al., 2018*; *Han et al., 2019*; *Cusack et al., 2020*; *Day, 2017*). The methylation of the fifth carbon (C5) of the cytosine ring to yield 5-methyl-2′-deoxycytidine (5mdC) was the first nucleotide modification to be discovered (*Hotchkiss, 1948*) and has remained the most intensively studied (*Umer and Herceg, 2013*; *Smith and Meissner, 2013*). 5mdC can be enzymatically oxidized into 5-hydroxymethyl-2′-deoxycytidine (5hmdC) and further into 5-formyl-2′-deoxycytidine (fdC) and 5-carboxyl-2′-deoxycytidine (cadC) (*Hu et al., 2015*; *Ito et al., 2011*; *Tahiliani et al., 2009*). Although these modifications have been described as transient intermediates of 5mdC demethylation, at least one (5hmdC) has been found to accumulate in the mammalian brain, specifically in the large Purkinje neurons, indicating a regulatory function (*Kriaucionis and Heintz, 2009*). N4-methyl-2′-deoxycytidine (4mdC), found in bacteria, is yet another form of cytosine modification (*Janulaitis et al., 1983*; *Ehrlich et al., 1987*). Cytosine thus exists in multiple chemical states (dC, 5mdC, 5hmdC, fdC, cadC, 4mdC, as well as the rare 4,5-dimethyl-2′-deoxycytidine [4,5dmdC]) (*Umer and Herceg, 2013*; *Klimasauskas et al., 2002*). Another important modification is the N6 methylation of adenine. N6-methyl-2′-deoxyadenosine (N6mdA) was initially discovered in bacterial genomes (*Dunn and Smith, 1955*) and later also in archaea, plants, and nematodes (*Couturier and Lindås, 2018*; *Liang et al., 2018*). Although N6mdA is not essential in microbial model organisms, this modification has been increasingly associated with functions that promote virulence or to counteract viral DNA integration (*Heusipp et al., 2007*; *O'Brown and Greer, 2016*). Indeed, it seems likely that DNA modifications play different roles in different species, as indicated by the varying amounts of DNA modifications across model organisms. For instance, *Arabidopsis thaliana* has orders of magnitude higher levels 5mdC compared to the dominant insect model *Drosophila melanogaster,* while the dominant yeast model organism *Saccharomyces cerevisiae* lacks this modification altogether (*Münzel et al., 2011*; *Capuano et al., 2014*).

Until recently, studying DNA modifications was technically challenging, information concerning their content and function was scarce for species other than model organisms, several crops, and humans. Moreover, it was rather difficult to translate the knowledge derived from those intensively studied species into a broader biological context. For instance, it is hard to judge from the current literature if the low amount of DNA modifications in laboratory yeast and *D. melanogaster*, or the high amount in *A. thaliana* (*Capuano et al., 2014*), represent the rule or the exception in their respective phylogenetic group without a broader multi-species dataset for comparison.

In addition to the position-specific information provided by sequencing technologies (*Chen et al., 2020*; *Liu et al., 2019*), global quantities of DNA modifications are required to obtain a complete picture about their function and metabolism. For instance, quantitative values are required to determine activity of the biochemical pathways that modify nucleic acids. Moreover, there are roles of DNA modifications that do not necessarily depend on their specific location in the genome, like in anti-viral immunity. Also, there might be relationships between different modifications that depend on their chemistry rather than their function. Last but not least, absolute concentrations can help to normalize the values as provided by sequencing technologies and to assess their false positive and false negative rates. We and others *Capuano et al., 2014*; *Le et al., 2011*; *Chowdhury et al., 2017*; *Tang et al., 2015*; *Chilakala et al., 2019*; *Gosselt et al., 2019* have shown previously that targeted mass spectrometry is an ideal technology to determine absolute quantities of DNA modifications, specifically, if they are low abundant and in the noise range of sequencing technologies. Mass spectrometry further is suitable for studying poorly characterized species, as no prior knowledge about the genome is required for data analysis. Aside from that, targeted mass spectrometry is economical, with running costs per sample amounting to single-digit dollars. For these reasons, mass-spectrometric quantification is well suited for identifying interesting patterns in the amount and relative abundances of DNA modifications, specifically within understudied species.

**Table 1.** Concentrations of pure nucleoside standards and their sources.

| Molecule: vendor/code | Pure stock concentration (µM) | Pool concentration (µM) |
|---|---|---|
| 2dC: Sigma/D3897-100MG | 5,000 | 100 |
| 5hmdC: Berry and Associates/PY7588 | 0.5 | 0.04 |
| 5mdC: Santa Cruz/ sc-278256 | 100 | 0.02 |
| cadC: Berry and Associates/PY7593 | 0.5 | 0.02 |
| dA: Sigma/D7400-250MG | 5,000 | 100 |
| dG: Sigma/854999 | 5,000 | 100 |
| fdC: Berry and Associates/PY 7589 | 0.5 | 0.02 |
| N6mdA: Alfa Aesar/ J64961 | 0.5 | 0.02 |
| T: Sigma/89270–1G | 5,000 | 100 |

# Results and discussion

## Global quantification of a panel of enzymatic DNA modification using liquid chromatography/multiple reaction monitoring

In order to quantify the global levels of multiple enzymatic DNA modifications in a single analysis, we expanded a previous method based on liquid chromatography-multiple reaction monitoring (LC-MRM) and designed for the quantification of 5mdC (*Tsuji et al., 2014*). This method is characterized by a sensitivity down to attomoles and a broad dynamic range, and discriminates between RNA and DNA modifications, clarifying the previously debated content of 5mdC in several yeast species (*Capuano et al., 2014*). In this method, isolated DNA is first enzymatically digested to obtain the corresponding nucleosides using a nuclease enzyme mixture (DNA Degradase Plus, Zymo Research). The resulting digest is directly analyzed by a targeted assay using LC-MRM using a triple quadrupole (QQQ) mass spectrometer. Distinguishing the nucleosides arising from a DNA monomer from a potentially co-purified RNA monomer occurs on the basis of the precursor mass difference of the sugar moiety. Such a strategy ensures the measured nucleosides are free from RNA contamination as many base modifications are also present in RNA (*Capuano et al., 2014*; *Tsuji et al., 2014*). For quantifying other DNA modifications, namely 5hmdC, N6mdA, cadC, and fdC, we obtained synthetic standards for these molecules and optimized the instrumental and chromatography parameters accordingly (*Tables 1 and 2*; *Figure 1—figure supplement 1*). Moreover, we supplemented the method by a neutral loss scan as a strategy to confirm the MRM results, as well as to detect additional modifications such as 4mdC, that were not included among the standards. Combined with the high sensitivity offered by a triple quadrupole mass spectrometer (Agilent 6470), we were able to achieve detection limits in picomolar ranges (*Figure 1A*).

**Table 2.** Retention times and transitions for nucleosides analyzed.

| Molecule | Precursor ion | Qualifier Product ion | Quantifier Product ion | Retention time (min) |
|---|---|---|---|---|
| 2dC | 228.1 | 95 | 112.0 | 4.362 |
| cadC | 272.0 | 137.9 | 155.9 | 5.193 |
| 5mdC | 242.0 | 108.6 | 126.0 | 5.130 |
| dG | 268.1 | – | 151.9 | 7.546 |
| fdC | 256.0 | 97 | 139.9 | 7.868 |
| dA | 252.1 | – | 136.0 | 8.128 |
| T | 243.1 | 54.1 | 126.9 | 8.349 |
| 5hmdC | 258.0 | 141.9 | 81.1 | 10.585 |
| N6mdA | 266.3 | 117 | 150.0 | 11.391 |

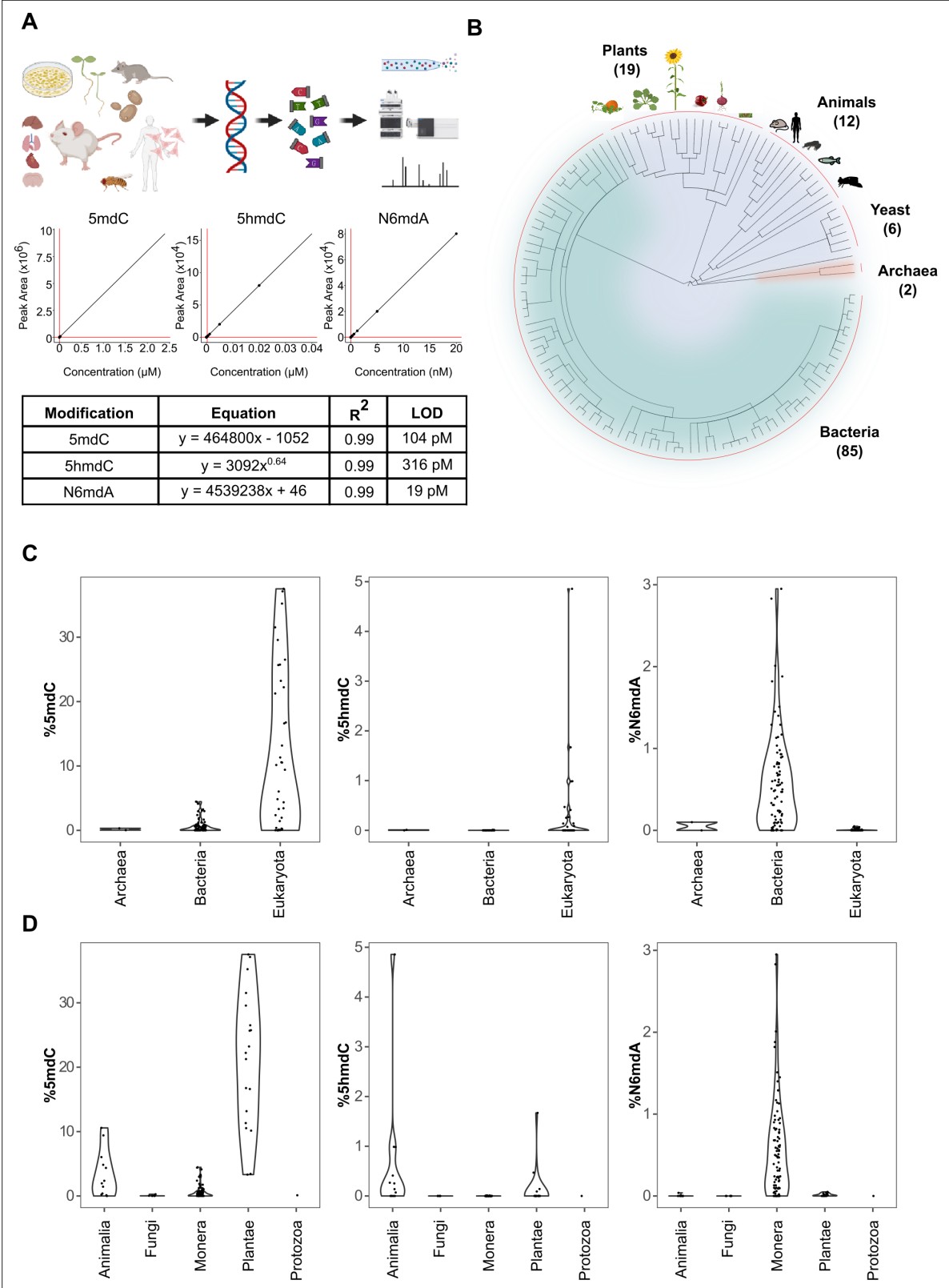

**Figure 1.** Quantification of DNA modifications across species.

(**A**) Multiplex analysis of various genomic DNA modifications using liquid chromatography-multiple reaction monitoring following enzymatic digestion of DNA. The regression curves and limit of detection (LOD) for modifications 5mdC, 5hmdC, and N6mdA are represented. Although our method also quantifies cadC and fdC, we did not detect significant concentrations of these in any of the measured samples; these modifications were hence

*Figure 1 continued on next page*

*Figure 1 continued*

omitted from the graphical illustrations. (**B**) A total of 286 tissue samples from 124 species were analyzed in the present study: 19 species from plants, 12 from animals, 6 from yeast, 2 from archaea, and 85 from bacteria. (**C–D**) Distribution of 5mdC, 5hmdC, and N6mdA across (**C**) archaeal, bacterial, and eukaryotic domains, and (**D**) animal, fungi, monera, plant, and protozoan kingdoms. The values depict percentage of cytosine residues bearing either methyl (%5mdC) or hydroxymethyl (%5hmdC) modification and percentage of adenine residues bearing methyl modification (N6mdA). Percentage modifications were calculated as ratio of modified cytosine residue and guanosine for 5mdC and 5hmdC; and ratio of modified adenine residue and thymine for N6mdA. The limits of detection for 5mdC, 5hmdC, and N6mdA are 4.6 nM, 320 pM, and 19 pM, respectively.

The online version of this article includes the following figure supplement(s) for figure 1:

**Figure supplement 1.** Representative extracted ion chromatograms for the standards and QC samples from different sample types.

**Figure supplement 2.** Neutral loss chromatograms corresponding to the transition 242->126 for the loss of ribose sugar moiety (M=116).

**Figure supplement 3.** The variation observed for samples measured in replicates.

**Figure supplement 4.** The variation of the three modifications 5mdC, 5hmdC, and N6mdA in *Drosophila melanogaster*.

Upon setting up the method, we sampled cells or tissues for a large number of species across the three domains of life. Because our method does not include any amplification steps and detects modifications on the DNA directly, it requires clean DNA at microgram levels, at least for the detection of the lowly concentrated DNA modifications. Unfortunately for some rare specimens, we only had limited sample amounts, and in many cases, standard DNA preparation protocols did not yield DNA of sufficient quality or concentration for our assay. However, by combining different protocols and sources, we were able to obtain clean DNA at microgram levels for 286 distinct tissues. To isolate DNA, we employed mostly a spin-column kit (Genomic-tip 20/G, Qiagen) which is chemically mild to DNA, and avoided strategies that involve the use of oxidants and reactive chemicals. However, for plant species, due to their biochemical composition, we were forced to use phenol–chloroform extraction to obtain sufficient quantities of DNA. In such cases, reagents like β-mercaptoethanol (2-sulfanylethan-1-ol) were included to keep DNA damage to a minimum during the extraction. The obtained DNAs were from 124 different species, including 85 bacterial species, 6 yeast species, 2 archeal species, 19 plant species, and 18 tissue and cell-culture samples from multiple animal species, including human and mouse. The collection included both the typical model organisms, and specifically for bacteria, vertebrates, and plants we included a significant number of species that have been barely characterized at the molecular level so far (*Figure 1B*). Furthermore, for a number of vertebrates, including human, the model organisms mouse (*Mus musculus*), African clawed frog (*Xenopus laevis*), but also for some less studied species, the opossum (*Monodelphis domestica*), the Alpine marmot (*Marmota marmota*), and the Oman garra (*Garra barreimiae*), we obtained DNA from multiple tissues and/or cell lines in order to quantify tissue differences in the absolute DNA modification content. For plants, we focused on seedlings that were germinated in the lab (*Varma and Calvani, 2022*). The seedlings not only allowed for efficient DNA extraction, which can be hampered by high concentrations of plant polymers in fully differentiated plant tissues, but also for direct comparison between the plants at a similar developmental stage. Multiple species were analyzed in replicates to identify the extent of variation in the analytical technique which revealed reasonably consistent values for modifications measured across different species (*Figure 1—figure supplement 2*).

## While multiple lower eukaryotes lack DNA modifications, N6mdA dominates in bacteria, and 5mdC is the dominating DNA modification across higher eukaryotes

Our results reveal major differences in the nature and global concentration of DNA modifications when comparing the domains of life (*Figure 1C, D*). First, despite the broad coverage, high sensitivity, and precision of our method, we did not detect significant levels of fdC and cadC in any of the genomes measured (limits of detection were 238 pM and 251 pM, respectively). These oxidized forms of 5-methyl-2′-deoxycytidine have been associated with the degradation of 5mdC (*Ito et al., 2011*), and according to our results they seem to remain undetectable across species as they are known to be labile and do not accumulate to significant, genome-wide-scale levels. In addition, neutral loss scans conducted in parallel, confirmed the picture that across species, only 5mdC, 4mdC, 5hmdC, and 6mdA reached notable concentrations on the genome-wide level. A notable exception was that we detected hardly any of these DNA modifications in the unicellular fungi studied (*Supplementary file*

*1*). Hence it is not merely 5mdC (*Capuano et al., 2014*; *Binz et al., 2018*; *Nai et al., 2020*), but also its oxidized form 5hmdC along with N6mdA that are very low if not absent in typical yeast species. It is interesting in this context that the insects *Trichoplusia ni*, *Spodoptera frugiperda*, and *D. melano-gaster* (*Supplementary file 1*) all had DNA modifications, but also at much lower levels compared to both, higher organisms but also bacteria. Indeed, the fruit fly *D. melanogaster* has so far been consid-ered an unusual case among the laboratory model organisms, as it contains only trace amounts, if any, of cytosine methylation (*Capuano et al., 2014*; *Lyko et al., 2000*; *Zhang et al., 2015*), but our data suggests this picture could be common to insects and other lower eukaryotes.

The presence of other DNA modifications in *D. melanogaster* like N6mdA has also been contested due to the presence of an appreciable gut microbiome, which could confound the results (*O'Brown et al., 2019*). We assessed this situation, comparing the genomic DNA obtained from fruit flies that possessed a functioning gut microbiome vs. ones grown under germ-free conditions. N6mdA was also detected in germ-free *D. melanogaster* (~0.04%, *Figure 1—figure supplement 3*). In a recent study comparing DNA adenine methylation levels in multiple eukaryotic species, the bacterial contami-nation affected the N6mdA measurements. However, it was possible to distinguish the N6mdA in *D. melanogaster* tissue from microbial contamination using quantitative deconvolution (*Kong et al., 2022*). While the adult *D. melanogaster* contained methylated adenine as a DNA building block, ovarian cells collected from two moth species (*T. ni* and *S. frugiperda*) principally contained methyl-ated cytosine as the preferred base modification (0.2 and 0.1%, respectively).

What conclusions can be drawn from the low concentrations of DNA modifications in yeasts and insects? First, these results support the notion that enzymatic DNA modifications are not universal, which could have peculiar evolutionary consequences. Studies in yeast have concluded that DNA modifications could have been specifically lost during yeast evolution (*Bhattacharyya et al., 2020*). However, our result that insects can have similarly low DNA modification levels raises another possi-bility that DNA modifications could have evolved in higher eukaryotes and bacteria, after yeasts and insects branching off. As a rule, most genomes contained a single modification type that did pass the limit of detection of the highly sensitive method. Some exceptions to this were, however, encountered. A subset of the eukaryotes and a subset of prokaryotic species contained low concentrations also of a second modification, which could be either 5mdC, N6mdA, or 5hmdC (*Figure 2*, *Figure 2—figure supplement 1*). For instance, *Diplotaxis tenuifolia* had low amounts of N6mdA (0.1%, *Supplementary file 1*) next to high amounts of 5mdC. Notably, species that exhibited 5hmdC were also observed to

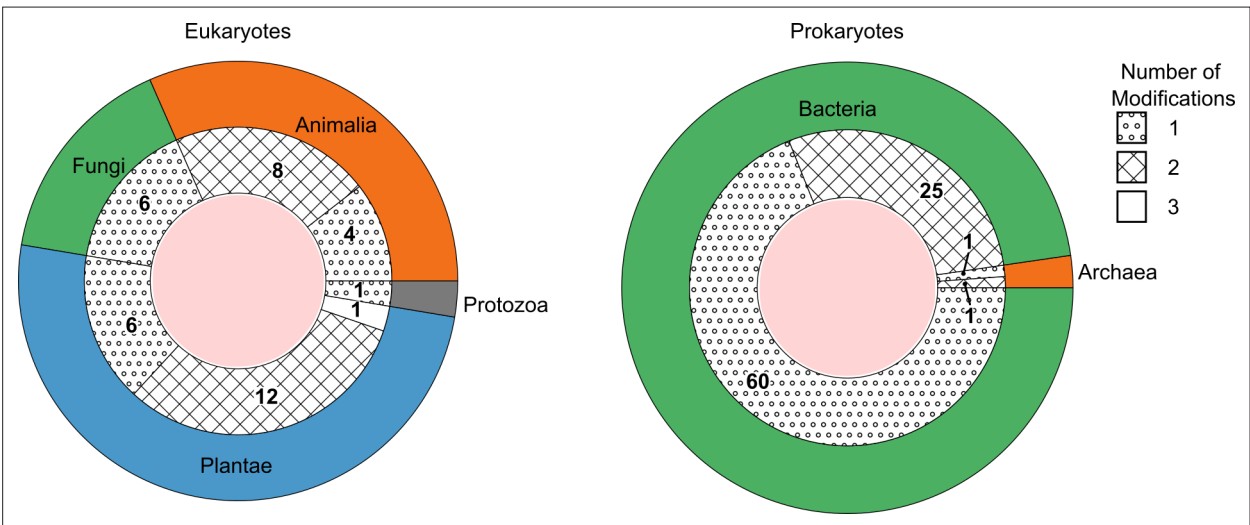

**Figure 2.** The number of species detected containing one, two, or three DNA modification types above picomolar detection limit, grouped as eukaryotes (left) and prokaryotes (right). The outer ring represents the kingdoms present within these domains. The groupings per number of modifications are shown as fill patterns on the inner ring, where dots represent species in which only one among 5mdC, 5hmdC, and N6mdA were found; crosses represent species bearing two modifications simultaneously; and no fill represents species carrying all three modifications.

The online version of this article includes the following figure supplement(s) for figure 2:

**Figure supplement 1.** Distribution of modifications across species.

contain its precursor 5mdC. Of particular interest was *Raphanus sativus*, which was the only species among those analyzed that possessed all the three modifications at detectable levels and in parallel. Among prokaryotes, we observed only cytosine and adenine methylation modifications, with 5hmdC entirely missing. Our study further featured two archeal genomes (*Sulfolobus acidocaldarius* and *Halobacterium salinarum*), which shared a similar level of the cytosine modification but differed in their levels of adenosine modification. While we detected N6mdA in *H. salinarum*, no adenosine modification was observed for *S. acidocaldarius* (*Supplementary file 1*).

## Tissue divergence of 5mdC concentrations in vertebrate and plant genomes

Among the DNA modifications, 5mdC had the highest abundance and was specifically abundant in plants. Most vertebrate genomes studied had a 5mdC content of around 5% (mean 4.66, SD 2.17) of the cytosine residues. Some species, including the model organisms *Danio rerio* and *X. laevis,* had higher levels consistent with early observations (*Colwell et al., 2018*). In plants, however, 5mdC concentrations of 10% (mean 20.34, SD 9.81) and higher were typical (*Figure 1D*). Extremely high values for cytosine methylation were observed in *Andropogon gerardii* and *Allium cepa*, where more than 35% of cytosines were methylated (*Figure 1D*, *Supplementary file 1*). As plants are known to possess polyploid genomes, high cytosine methylation values could be attributed to silencing of multiplied genes and the much larger non-functional parts of their genome (*Masterson, 1994*). Given that very low levels or no 5mdC were detected in yeast and insects, cytosine 5 methylation levels hence differ by several orders of magnitude within the eukaryotic kingdom.

In multicellular organisms, DNA modifications are important for development, and tissue differences between DNA modification patterns are observed (*De Bustos et al., 2009*; *He et al., 2020*; *Zhu et al., 2018*). Our data suggests that a change in the modification pattern or sequence context does not necessarily have a strong impact on the total concentrations of the DNA modifications, however. We analyzed spleen, muscle, lung, liver, kidney, heart, and CNS samples from five animal species, of which two are model organisms (*X. laevis*, *M. musculus),* and three non-model organisms (*G. barreimiae*, *M. domestica*, *M. marmota*). From *M. musculus* we further examined tissues from multiple inbred laboratory lines: BALB/c, FVB/N, Hsd/Ola/MF1, B6SJL/CD451/CD452, BALB/cAnN, 129S8, and F1/CBAxB6. In parallel, we analyzed multiple human cell lines (*Supplementary file 1*). The obtained data was consistent, in the sense that the values for 5mdC levels were highly similar, as long as the tissues were derived from the same species (*Figure 3A*, left). For instance, most tissues in *G. barreimiae*, *M. marmota*, and *M. musculus* tissues had 5mdC levels of around 5–6% (*Figure 3A*). Between the different mouse lines, there were no significant differences in 5mdC levels (*Supplementary file 1*). We noted, however, some small but notable differences between specific tissues. Heart tissue presented a broad cytosine methylation level and brain tissue had a higher median value for percentage methylation compared to other tissues (5.3 vs. 4.9%; *Figure 3B*). We then tested whether different nutritional conditions would change the picture. Therefore, we grew a commonly used mammalian cell line (HeLa) under different growth conditions. The different growth conditions affected 5mdC levels, and the detected differences were in a similar magnitude as the small differences detected between tissues (*Figure 3—figure supplement 1*).

Overall, 5mdC concentrations in *M. domestica* (opossum) and *X. laevis*, respectively, were different to the aforementioned species. In opossum, we detected much lower levels (2%) of 5mdC in all tissues examined. Conversely, in *X. laevis*, all tissues had much higher concentrations (about 9.4%). Higher values in *X. laevis* could be attributed to the tetraploid genome of this species compared to its relative *X. tropicalis*, which is diploid (*Head et al., 2014*). However, also here, in both cases the tissue differences in the 5mdC concentrations were minimal, at least when compared to the differences that exist between species. Although we tested fewer cases in plants, our data suggest the situation could be similar there too. We tested different tissues (roots, leaf, stem, and seed cotyledon) from *Phaseolus vulgaris* and obtained consistently high (16.7%) 5mdC concentrations in all measured tissues (*Supplementary file 1*). Hence, the several tissues examined from animal species, cell lines, and *P. vulgaris* provided a largely consistent picture: in a given organism, several tissues exhibit similar levels of 5mdC, and, that within-tissue differences are typically smaller compared to the differences that can be detected between species.

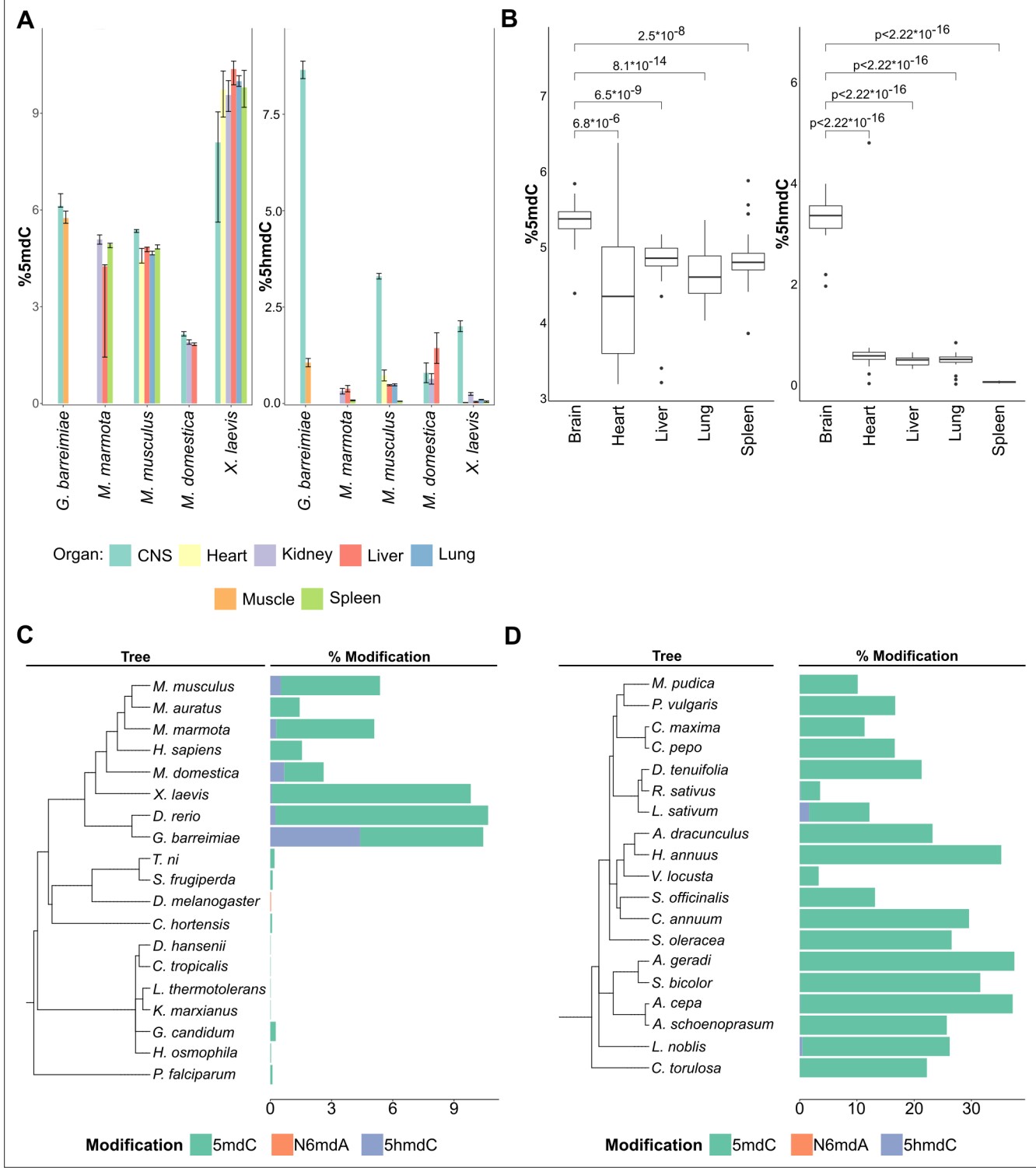

**Figure 3.** Distribution of DNA modifications in eukaryotes.

(**A**) The concentration of 5-methyl deoxycytidine (left) and 5-hydroxymethyl deoxycytidine (right) in different vertebrate genomes. n = 4 for *G. barreimiae*, *M. marmota*, *M. musculus*, *X. laevis* and n=3 for *M. domestica*. (**B**) Distribution of 5-methyl deoxycytidine (left) and 5-hydroxymethyl deoxycytidine (right) in different mouse tissues (n=5). Variations in percentage modification across different (**C**) non-plant eukaryotes including representatives from vertebrates like mammals, amphibians, and fish, invertebrates like insects and mollusks, and unicellular fungi and protozoa (**D**) plants species comprising both gymnosperms and angiosperms.

The online version of this article includes the following figure supplement(s) for figure 3:

*Figure 3 continued on next page*

*Figure 3 continued*

**Figure supplement 1.** Variation in % modification (as $\log_{10}$) in HeLa cells cultured under different growth environments: standard media, media with no penicillin/streptomycin, media with 10x penicillin/streptomycin concentration, incubation at 40°C, treatment with 2.5 mM DTT, treatment with 200 µM $H_2O_2$, and interferon gamma treatment.

## Tissue specificity of 5-hydroxymethyl deoxycytidine in the vertebrate CNS

Tissue specificity was, however, detected for another modification, 5hmdC. Indeed, 5hmdC was previously discovered in mammalian brain tissue, where it is formed via oxidation of 5mdC by TET enzymes (*Tahiliani et al., 2009*; *Globisch et al., 2010*). Our dataset shows that 5hmdC is detected in a broad range of vertebrate tissues except for spleen, but reaches significantly higher concentrations specifically in samples from the CNS. Although the spleen tissues had similar 5mdC levels as other mouse tissues, 5hmdC was not detected in these tissues (*Figure 3B*). Interestingly, our data reveals that the highest 5hmdC levels were not detected in the mammalian brain, but in the fish *G. barreimiae.* were levels could reach up to 8% of cytosine residues Although lower compared to *G. barreimiae*, mammals *M. musculus* (3.3%), and amphibian *X. laevis* (2%) still had high levels of 5hmdC specifically in brain tissue relative to other tissues in those organisms (*Figure 3A*, right). An interesting exception was in opossum, the only vertebrate species analyzed, in which 5hmdC levels were not higher in the brain compared to peripheral tissue.

Apart from vertebrates, 5hmdC was also observed in *A. thaliana* and *Oryza sativa* (*Mahmood and Dunwell, 2019*). Our data shows that the presence of 5hmdC is by no means universal in plants, indeed, we did not detect it in the majority of plant samples. Nonetheless, our data adds several species (*A. cepa, Laurus nobilis, Lepidium sativum*, and *R. sativus*) in which we confirmed low concentrations of 5hmdC. Furthermore, we did not detect 5hmdC in any of the bacterial or fungal genomes analyzed. Our results support the fact that the modification of 5hmdC is more widespread in biological systems than previously assumed, but quantities above picomolar-levels being not detected in any bacteria, yeasts and many tissues from higher organisms implies 5hmdC is not universal or specific to any part of the phylogenetic tree.

## Variations in DNA modification across different bacterial species

In prokaryotes, high amounts of DNA modifications all concerned N6mdA, with the highest levels detected in *Mobiluncus curtisii* (~1.4%) and *Moorell thermoacetica* (~1.1%). In total, the prokaryotic genomes hence contained higher amounts of DNA modifications compared to lower eukaryotes such as yeasts and insects, but lower amounts of DNA modifications compared to higher eukaryotes—plants and vertebrates in particular. Typical bacterial species contain only one type of modification—mostly N6mdA (*Figure 4A*). Our data reveals some exceptions. Certain genera such as *Campylobacter* contain trace quantities of 5mdC (<0.1%) next to the dominating N6mdA modification (*Supplementary file 2*). In general, the observed trend was that the occurrence of one type of modification limits the occurrence of the other. For instance, *M. curtisii* with ~1.4% of its adenine residues methylated shows only 0.3% 5mdC, while *Sebaldella termitidis*, with unusually high cytosine methylation (~2.4%), has only 0.1% of its adenines methylated. Interestingly, we observed that median values for 5mdC dominate over N6mdA in those bacteria that colonize or enter mutualistic relationships with higher eukaryote species that carry 5mdC as their main modification (*Figure 4—figure supplement 1*, *Supplementary file 2*). This included the genus *Neisseria*, mucosal-surface-colonizing bacteria, which showed 1.4 and 2% (*Neisseria gonorrhoeae, Neisseria lactamica,* respectively) of cytosine residues were methylated while containing only <0.3% N6mdA, and *Faecalicoccus pleomorphus* and *Bifidobacterium adolescentis*, with >1.5% of 5mdC without any detectable levels of N6mdA modification. Indeed, others made a similar observation in single-cell fungi. While the environmental yeasts studied herein and previously lacked any modifications (*Capuano et al., 2014*), the most frequent commensal yeast pathogen *Candida albicans* contained as sole yeast species 5mdC (*Mishra et al., 2011*). This result is interesting, because it could mean that host–pathogen interactions could select for similar DNA modifications in the pathogen as in the host. The study of future host–pathogen pairs is necessary to substantiate this observation and suggests that the picture about the functions of DNA

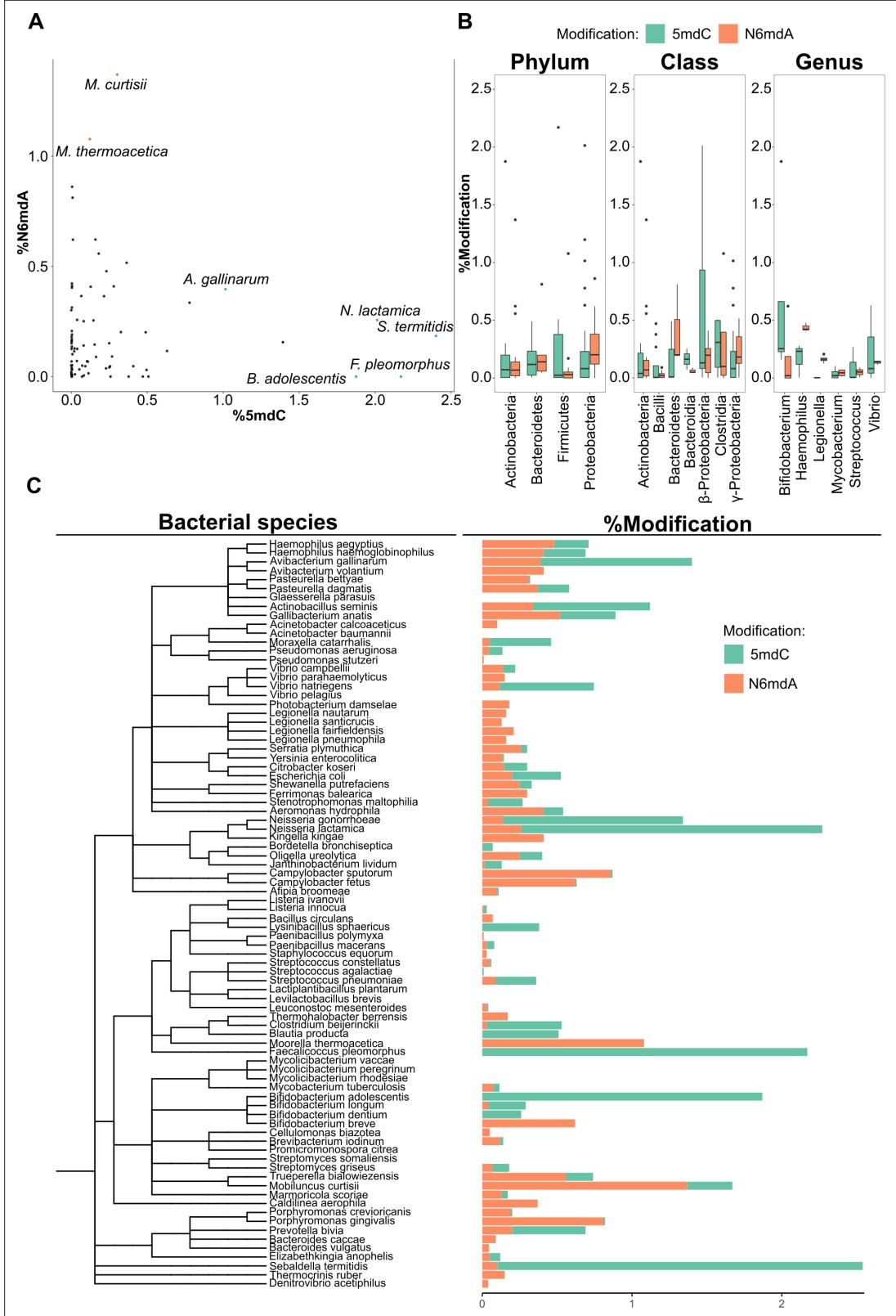

**Figure 4.** DNA modifications in bacteria. (**A**) Percentage of cytidine methylated against the percentage of adenine methylated in bacterial species. (**B**) Variation of % 5-methyl deoxycytidine and % N6-methyl deoxyadenosine among taxonomic divisions: phylum, class, and genus. One-way ANOVA, p-values for phylum, class, and genus are 0.017, 7×10⁻⁴, and 0.16, respectively. (**C**) Distribution of 5mdC and N6mdA among 85 bacterial species depicted

*Figure 4 continued on next page*

*Figure 4 continued*
together with their phylogenetic relationships. Percentage modifications are calculated as ratio of modified cytosine residue and guanosine for 5mdC and 5hmdC; and ratio of modified adenine residue and thymine for N6mdA.

The online version of this article includes the following figure supplement(s) for figure 4:

**Figure supplement 1.** Variation of 5mdC and N6mdA among hosted and free-living bacteria.

**Figure supplement 2.** Variation of 5mdC and N6mdA according to genome length and % GC content.

modifications in prokaryotes is incomplete; 5mdC has thus far not been associated with function in pathogen immunity.

Having analyzed 85 species, we were able to ask if bacterial species with a close evolutionary relationship or similar habitat or genome properties also have a more similar modification makeup. We did not detect any relationship between nature and level of modification with genome size or GC content (*Figure 4—figure supplement 2*). Similarly, we detected no significant correlation between factors such as pathogenicity, temperature of growth, or tolerance to oxygen and the amount of modifications per unit genome size (not shown). We did, however, observe obvious patterns at the different taxonomic levels once we grouped the different bacterial strains according to phylum, class, and genus. Similarities are detected at the genus level (*Figure 4B, C*). Members of the same genus often displayed similar values for a given modification. For example, species of the *Vibrio* genus presented similar quantities of N6mdA. At the class level, we observed trends between the different classes and the amount of modification. α- and γ-Proteobacteria had the highest N6mdA content among different classes present while bacteroidetes presented with more cytidine methylation than adenosine methylation. At the phylum level, the patterns were more prominent in Proteobacteria, containing more N6mdA than 5mdC, while a reverse trend of more 5mdC than N6mdA was observed for Bacteroidetes and Firmicutes. Finally, we also observed a third modification, 4mdC, to be frequent in prokaryotes. 4mdC was detected in tandem with 5mdC as a second modification in *Shewanella putrefaciens*, *Stenotrophomonas maltophilia*, *Bifidobacterium dentium*, *M. curtisii*, and *Gallibacterium anatis* (*Figure 1—figure supplement 2*, not quantified).

Although the exact mode of inheritance could not be inferred from the present study, it is worth pointing out that the vastly different amounts of DNA modifications also indicate differences in the way they are inherited. It is plausible that the activities and specificities of DNA methylases and demethylases differ between species with high or low amounts of global DNA modifications. Combined, these results suggest that differences in the modifications do not reflect basic structural genome features such as size or GC content, but rather show that more closely evolutionarily related species have higher similarities in DNA modification implying gene drift and gene function are key drivers in the evolution of DNA modifications.

## Materials and methods

For a description about the sources of samples and their extraction, please refer to 'Supplementary Information for sample sources: Global analysis of cytosine and adenine DNA modifications across the tree of life' (*Varma and Calvani, 2022*).

### DNA extraction

DNA extracts were treated with RNase A (VWR, Cat.No. E866-5ML) at 37°C for 45 min, and DNA purification was performed using QIAGEN Genomic tip-20/G according to the manufacturer's instructions. Purified DNA was precipitated with isopropanol, washed with 70% ethanol, and resuspended in 10 mM Tris-HCl, pH 8.0. Quantification was done using a dsDNA BR Assay Kit (Qubit). The DNA sample was then digested into corresponding nucleosides using DNA Degradase Plus (Zymo Research, E2020). 1 µg of DNA was treated with 5 U of DNA degradase at 37°C for 2 hr in a final volume of 25 µl and the reaction was inactivated by incubating the samples for 20 min at 70°C as described by the manufacturer. Calibration standards were prepared in 1:4:4:2:2:2:2:4:4:4:4:4:4:10 serial dilutions from a standard stock that was prepared as per *Table 1* and stored at –80°C.

## Instrumentation

The samples were diluted 1:1 with MeOH 10% (v/v) containing 0.2% formic acid, and 10 µl corresponding to 200 ng of gDNA were injected onto a liquid chromatography system equipped with reverse phase Acquity UPLC HSS T3 column, 100 Å, 1.8 µm, 2.1 mm × 150 mm (Waters), column temperature 25°C and flow rate of 0.2 ml/min. Mobile phase A: 0.1% formic acid +10 mM ammonium formate in water, mobile phase B: 0.1% formic acid +10 mM ammonium formate in methanol. Gradient for elution was started from 5% mobile phase B to 35% B over 11.5 min followed by sharply increasing to 80% over the next 1.5 min. The gradient was held at 80% B for 2 min, lowered to the starting gradient over 1 min, and equilibrated for 6.5 min. Total length was 22.5 min.

The eluent was directed to an electrospray ion source connected to a triple quadrupole mass spectrometer (Agilent 6470 QQQ) equipped with an Agilent Jet stream source, operating in positive mode. The ESI source settings were: gas temperature: 300°C; gas flow: 6.4 l/min; nebulizer: 50 psi; sheath gas heater: 350°C; sheath gas flow: 7 l/min; capillary: 2000 V. The transitions monitored for MRM experiments are listed in *Table 2*.

For neutral loss experiments, the samples were injected as per the same LC parameters used for the MRM experiment while the mass spectrometer was set to a scan type of neutral loss (M=116 Da) while scanning the quadrupoles from 230 to 250 Da. The scan time was 1000 with a step size of 0.05 amu and the values for Fragmentor, collision energy, and cell accelerator voltage were 73, 8, and 5, respectively. 4mdC was detected as the second peak in the neutral loss ($\Delta$ = 118) chromatogram corresponding to parent ion 242 Da.

## Data processing and analysis

Peak areas were extracted and integrated using MassHunter for QQQ to obtain the concentrations after applying the necessary limits of quantification. Subsequent processing for batch-to-batch variation and technical outlier removal were carried out using R or Python. A single reference mouse DNA sample was included in every measured batch to monitor batch-to-batch variation. Median-value-based normalization of the reference mouse samples was used to obtain the correction factor with which the corresponding batch was corrected. The results are depicted as percentage modification with respect to dG (for 5mdC and 5hmdC) and T (for N6mdA). The phylogenetic clustering was carried out using a newick file generated using NCBI Taxonomy (PhyloT) and the ggtree package (*Yu, 2020*). Features of bacteria were retrieved from the bacterial metadatabase BacDive (http://bacdive.dsmz. de, accessed April 14, 2020; *Reimer et al., 2019*).

## Acknowledgements

We thank Biological Research Facility at Francis Crick Institute for *Mus musculus*, *Danio rerio*, *Xenopus laevis*, samples, Bryony Lee (Turner lab, The Francis Crick Institute) for opossum samples, Annick Sawala (Gould Lab) for *Drosophila* samples, Cell Services (The Francis Crick Institute) for animal cell lines, National Yeast Collection for yeast samples, Felix Forest (Kew Gardens) and Nell Jones (Chelsea Physic Garden) for plant samples, Barbara Tautsher, Elisabeth Haring, Luise Kruckenhauser, for *Cepaea hortensis* and *Garra barreimiae* samples (Natural History Museum of Vienna), Florian Winkler, Heinrich Aukenthaler, Erhard Seehauser, and Gottfried Hopfgartner (Forestry and Hunting Authorities South Tyrol, or Jagdrevier Mauls, Bolzano Province, Italy) for their support in obtaining tissue samples from alpine marmot in their wild habitats of Mauls and Gsies (Italy). We thank Christiane Kilian and Daniela Ludwig (Charité Universitätsmedizin Berlin) for yeast, plant and cell line samples. We thank Skirmantas Kriaucionis, Rob Klose, Julian Parkhill, Benjamin Heineike and Hezi Tenenboim for providing feedback on our manuscript. This work was supported by the Francis Crick Institute which receives its core funding from Cancer Research UK (FC001134), the UK Medical Research Council (FC001134), and the Wellcome Trust (FC001134), and received specific support from the Wellcome Trust (200829/Z/16/Z, 101503/Z/13/Z) and the German Ministry of Education and Research (BMBF) as part of the National Research Node "Mass spectrometry in Systems Medicine (MSCoresys)", under grant agreement 031L0220A.

## Additional information

### Funding

| Funder | Grant reference number | Author |
|---|---|---|
| Cancer Research UK | FC001134 | Markus Ralser |
| Medical Research Council | FC001134 | Markus Ralser |
| Wellcome Trust | FC001134 | Markus Ralser |
| Federal Ministry of Education and Research (BMBF) | 031L0220 | Markus Ralser |
| Wellcome Trust | 200829/Z/16/Z | Markus Ralser |
| Wellcome Trust | 101503/Z/13/Z | Markus Ralser |

The funders had no role in study design, data collection and interpretation, or the decision to submit the work for publication. For the purpose of Open Access, the authors have applied a CC BY public copyright license to any Author Accepted Manuscript version arising from this submission.

### Author contributions

Sreejith Jayasree Varma, Data curation, Formal analysis, Visualization, Writing - original draft; Enrica Calvani, Conceptualization, Data curation, Investigation, Methodology; Nana-Maria Grüning, Visualization, Writing – review and editing; Christoph B Messner, Michael Mülleder, Methodology, Resources; Nicholas Grayson, Resources; Floriana Capuano, Conceptualization; Markus Ralser, Conceptualization, Funding acquisition, Supervision, Writing – review and editing

### Author ORCIDs

Sreejith Jayasree Varma http://orcid.org/0000-0002-1669-2254
Nicholas Grayson http://orcid.org/0000-0002-9998-6783
Markus Ralser http://orcid.org/0000-0001-9535-7413

### Decision letter and Author response

Decision letter https://doi.org/10.7554/eLife.81002.sa1
Author response https://doi.org/10.7554/eLife.81002.sa2

## Additional files

### Supplementary files
• MDAR checklist

• Supplementary file 1. Percentages of 5mdC (wrt dG), 5hmdC (wrt dG), and N6mdA (wrt T) of different samples measured*. The values are presented as % modification: %5mdC as (5mdC/dG)*100, %5hmdC as (5hmdC/dG)*100 and %N6mdA as (N6mdA/T)*100.* Percentages are NOT normalized to genome size.

• Supplementary file 2. Percentages of 5mdC (wrt dG), 5hmdC (wrt dG), and N6mdA (wrt T) of different bacterial samples measured and their habitats*. Information regarding bacterial habitats was retrieved from the bacterial metadatabase BacDive (http://bacdive.dsmz.de, Accessed 14 April, 2020) (*Reimer et al., 2019*).*Percentages are normalized to genome size.

### Data availability

All data is available as Supplementary Materials. The sample source information is provided as a separate document (Varma, Sreejith; Calvani, Enrica (2022)), Supplementary Information for sample sources: Global analysis of cytosine and adenine DNA modifications across the tree of life, Mendeley Data, V1, doi: (https://doi.org/10.17632/jnbn8c2phv.1).

The following dataset was generated:

| Author(s) | Year | Dataset title | Dataset URL | Database and Identifier |
|---|---|---|---|---|
| Varma S, Calvani E | 2022 | Supplementary Information for sample sources: Global analysis of cytosine and adenine DNA modifications across the tree of life | https://doi.org/10.17632/jnbn8c2phv.1 | Mendeley Data, 10.17632/jnbn8c2phv.1 |

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
