## [Editor Report]

DNA methylation is an important mechanism to control gene expression, yet methods for quantitation of global DNA methylation analyses are limited. This work provides a new sensitive method for the quantitation of global DNA methylation and they apply this to over 100 species of eukaryotes and prokaryotes, finding interesting differences across species. This is a useful tool and resource for those interested in DNA methylation and evolution.

---

## [Decision Letter]

[Editors' note: this paper was reviewed by Review Commons.]

---

## [Author Response]

We would like to thank the reviewers for their valuable and constructive comments. We further would like to stare that we find the approach of Review Commons very refreshing. It’s great to receive comments on the science that is presented, without being judged about the suitability of the manuscript for a specific journal (and the cultural bias that has evolved around that).

Reviewer #1:Specific points:1. The authors observed a wide range of cytosine modification percentages across different living species, but have not really offered a plausible biological explanation of why that is. Could the authors provide some more speculation about the biological significance of this observation and species-dependent variations?

We thank the reviewer for their constructive feedback. The quantification of DNA modifications across many species does of course not provide a mechanistic explanation why the concentration of these DNA modifications has evolved so differently. Generally, it is however possible speculate about the key differentiators. It is hard to avoid noticing the most complex and large genomes, have high concentration of DNA cytosine methylation, while the most compact genomes, are bacterial genomes, possess with N6mdA. The fungal and insect genomes which are in between, are compact but possess the structure of a eukaryotic genome, are low in modifications in general. The key drivers are hence likely gene expression regulation in the complex genomes, and preventing elements that interfere with high compactness (like transposable elements or viral genome replication) in the bacterial genomes. (Page 8, first paragraph)

2. As a rather striking example (Figure 1D/D), why would the species of Eukaryota and Plantae have much higher frequencies of 5mdC as compared to the other species that were examined?

Not only do these species possess larger genomes, but they also contain much larger intergenic regions, more pseudogenes, and transposable elements. We hence speculate that one of the main use of DNA modifications is to be able to both to suppress the expression of non-functional genomic elements, and for gene expression regulation. (page 9 and 10)

3. As a conclusion, the authors mention "….related species have higher similarities in DNA modification suggests that gene drift and gene function are key drivers in the evolution of DNA modifications". Do the authors relate to a degree of "inheritance" of these modifications? This is in analogy to histone epigenetic modifications and is worth discussing also in this context.

We can derive conclusions from on the evolutionary relationships, but not conclude on the mode of inheritance from our comparative analytical study. But its worth speculating that the vastly different amount of DNA modifications also points to differences in the way they are inherited. It is plausible that the activities and specificities of DNA methylases and demethylases differs between species with high- our low amounts of global DNA modifications. We have hence expanded the section accordingly. (page 12, last paragraph).

4. The methodology or DNA detection by mass spectrometry is described in the Supplementary section, including Table S2 showing retention times and MRM transitions used by the authors. However, the authors do not show any examples of primary data, such as chromatography (e.g. TIC / EIC) profiles and mass spectra using standards and showing how they detected these modified nucleosides within the context of sample matrix. Examples reflecting the detection of each DNA modification should be provided and included.

We appreciate this suggestion. EICs for standards and sample matrix measurements from cell lines, bacteria, mice and plant samples are now included as Figure S1.

Reviewer #2:– The Title should reflect the fact that this study is mainly focused on 5mC, 5hmC and 6mA. The term DNA modifications is much wider than the modifications assessed by the authors (e.g. the authors do not attempt to analyse the content of 5hmU, 5gmC etc as well as DNA lesions such as 8oxoG in the corresponding genomes). Therefore, the current title is slightly misleading.

We appreciate the comment of the Reviewer. There are two aspects here. Indeed, we have measured, and screened, for a much broader set of modifications. The reason we have concentrated on 5mC, 5hmC and 6mA is however because these were the only modifications that we detected at significant concentrations in the samples. 8oxoG is an exception, it’s also present, but this modification has a very different biology, as an intermediate step of the excision repair pathway, and was omitted as it’s a stress signal, and hence strongly condition dependent. Also, the neutral loss scan experiments we conducted, confirmed, that in all species analyze*d,* only 5mC, 4mc 5hmC or 6mA reach more than trace levels of abundance across all species. We apologize that this important point situation was not sufficiently explained and have reworked abstract and rationale accordingly.

– The authors should clearly state what they mean by the 'modification percentage' in each figure/legend. E. g. 5hmC/C %, or 5mC/C percentage. This should be explicitly written in each panel.

We have now included the description for "modification percentage" for each figure.

– Slightly more speculation on the biological functions of the DNA modifications and their phylogenetic distribution would make the paper more interesting. In this regards the incorporation of a separate 'Discussion' section would be beneficial for the manuscript.

We agree, also reviewer #1 made that point. As both reviewers encourage us to become more speculative, we have now expanded the discussion with some suggestions and hopefully interesting speculations. (page 8, 9 and 10)

Reviewer #3:1. It would be relevant to conduct at least a subset of analyses as independent duplicates or triplicates in order to define the expected error margins.

We have conducted the analyses in triplicates (or more) in most instances, i.e. in all

those situations where we could obtain sufficient biological material. We have now included a figure (Figure S1) for species where measurements have been carried out in replicates. Generally, owing to the very high analytical precision of LC-SRM methods, the replicates are in excellent agreement.

2. It would be highly interesting to indicate LODs (or at least approximate LODs) in some of the figures, like 1C, D, 2A, B or 3A, B.

We have included the limits of detection for Figure 1-4.

3. I find the general concept of "presence" of a modification not convincing. This refers to Figure 2 and several occasions in the text, where it is said, that most species (and most bacteria) tend to "have" only one modification. Here the threshold between presence and absence is not properly defined. The statements also go against established knowledge telling us that many eukaryotes have 5mC and 5hmC (i.e. two modifications), and several bacteria do as well, like the well-know *E. coli*, which have the Dam and Dcm system generating 6mA and 5mC.

We apologize for the confusion, and the reviewer is of course right that we can and should be very specific here. By the usage "presence of a modification", we of meant to convey that the modification was detected above the (extremely sensitive) detection limit of our method. A modification not being detected does certainly not exclude the possibility of it being present at low levels- but owing due to the high sensitive and the data-independent nature of our approach – this can be only be trace quantities (just to note, we have far better detection limits as sequencing methods). We agree with the Reviewer that we have, and can, to be more precise in this respect, and have updated our text accordingly to include limits of detect with every figures.

4. 5hmC cannot exist without having 5mC, so I doubt if this concept overall makes sense.

Our data suggests the reviewer is right, but in theory, there was the remote possibility that there are species out there which convert all 5mC to 5hmdC. Our technology is agnostic to the order in which modifications are made or removed. Our data thus far supports the reviewer’s stance: In all cases where 5hmdC was detected, we detected also 5mdC consistently. (Table S3). (page 8, paragraph 2). The manuscript has been updated accordingly.

5. From a technical point of view, it would be relevant to document that distinction of 5mC and 4mC is really reliable.

We can distinguish the two modifications based on their chromatographic retention. Chromatograms for the neutral loss transition 242 -> 126 corresponding to the loss of ribose moiety (M=116) has been included along with authentic 5mdC TIC. To illustrate this, we are now including a chromatogram presenting an example of an organism containing 4mdC in majority(*Citrobacter koseri*) or an example of an organism containing 5mdC in majority 5mdC (*Listeria innocua*) or both (*Shewanalla putrefaciens*).

6. I wonder whether it would not be better to have a log scale y-axis in several of the figures.

We have considered the suggestion of the reviewer, for using log scale to present

the data. However, part of the value of our technology is that we can accurately quantify also low levels, and low differences, of the modifications and to distinguish closely related species. We believe log scale could make it difficult for readers to compare against many of the existing reports that do not use log scales. Rather we have tried to segregate species with similar magnitude together. We agree it’s a compromise, depending on whether one is more interested on the macroscopic rather than microscopic picture, the one or the other would be better.

7. Current literature has documented an import of modified nucleotides with the biochemical reagents. Can the authors exclude this and/or comment on this?

Indeed, the fact that artifacts of sample preparation does affect the values of modifications in the literature due to chemical reactions has also been a motivation for us in developing this analytical method, which is complementary to sequencing techniques. The reviewer is right that the chemistry does also not go away in LC-SRM technologies, although here in the problem is confined to chemical conversion of nucleotides, while we do not suffer the much bigger problem, that comes i.e. from the amplification of nucleic acids in several sequencing methods.

Indeed, the anger of artefacts was one reason to not include 8oxG as stress-sensitive modification in our study.

Further, our method avoids the usage of oxidants and other reactive reagents and instead used strategies that are much milder on the DNA (ex. spin column). For plant species, due to their biochemical composition, we were forced to use phenol-chloroform extraction to obtain enough DNA. But the inclusion of reagents like 2-thio ethanol, could keep this to a minimum. We discuss this now at much more depth in the manuscript, and added a caveat (page 6 paragraph 1).

8. Minor commentsThe sentence in the abstract regarding host-pathogen interactions is not backed up by data.

We agree in principle, this were only anecdotal observations, we had not enough host-pathogen interactions represented in our study to be able to claim this conclusion being broadly robust. We have removed the statement from the abstract, and backed down on the claim in the results, just mention it as a possibility that would be compatible from the few cases contained in our data in the results and Discussion section.

Introduction: Sequencing methods do provide quantitative methylation information, but only for individual sites not globally. This should be corrected.

The proposed correction has been implemented. We have used the sentence of the reviewer also updated the abstract, to highlight the complementarity.

Introduction: typo in consent, which should be content

The proposed correction has been implemented.

Figure 4 legend: …species displayed together with their phylogenetic tree.

The proposed correction has been implemented.

Reviewer #4:Major comments:– The manuscript is well written, but brief, in some cases, too brief. There needs to be a more robust discussion about the function of DNA methylation, what is already known about its function, patterning, levels across different kingdoms and phyla, and what this finding brings to our understating of epigenetics aside from cataloging the presence or absence of specific modifications.

We agree, indeed also Reviewer’s 1 and 2 encouraged us to be less neutral in presenting our results and encouraged us to speculate a little more. We have hence expanded the discussion.

– Some of the data are presented in a robust way while others are presented in a redundant or not linear way with scant detail or depth. More consistency in the way data is presented is warranted. For instance, Figure 2 is overly simplified, figure 3 lacks labels of the kingdoms and phyla represented, and there is no meaningful evolutionary distance assigned to may of the graphs.

We apologize for the situation but this is not our fault. The taxonomy in biological sciences is far from being ideal with different taxonomic classifications used by different communities (ex. phylum, class etc. are valid for prokaryotes and eukaryotes but not for plants which are described using clades). A solution, although not ideal, was to cluster the species based on their taxonomic proximity. We give however all results in the detailed supplementary tables (Table S3), so results from each species are well contained and accessible.

– Since most studies of DNA methylation include the number of modified C/ number of total C, the parallelism of this study with other data is difficult.

We are a bit confused by the reviewer’s comments. We have re-searched the literature. but could not reproduce that our study is replicating others. Most species contained in our dataset are indeed analyzed for their content of the DNA modifications for the first time. In other aspects, our study complements manuscripts that use sequencing technologies, with overall numbers.

– There is a glaring omission the data: since methylation is measured on cytosines, and, for 5methyl-cytosine typically in a CpG dyad, without knowing how CpG specifically and cytosine overall is present in these genomes, it is difficult to interpret. While this data cannot be obtained accurately in the absence of a reference genome, for those organisms where there is a reference, it should be included. This point should also be added to the introduction and discussion

The reviewer comment applied to the higher eukaryote genomes, several of which are included in our study. We agree with the reviewer that position specific information as obtained by DNA sequencing is essential to interpret the specific role of 5mC in these species, specifically in CpG islands, but there are other reasons for why one also needs the total numbers. (see above). We have revised the introduction and rationale accordingly, and hope that the complementary nature of our study is now clearer.

– It is not clear why the authors choose to represent the DNA modification values as a percentage of modified C or A over respectively G and T [as reported in the supplemental material (5mdC/dG or N6mdA/T)*100]. This analysis generated values of DNA methylation that are not apparently consistent with previous datasets (i.e human, mouse, zebrafish, etc.). It is difficult to understand the data of the new species analyzed and to make parallelism across those. The authors should consider using the total number of C or A recovered in the analysis of each sample and where possible compare and verify that this amount corresponds to the one annotated in the available genomes.

For most of the species studied in our manuscript, there are no analytically determined absolute concentrations of DNA modifications found in the literature. Furthermore, is difficult to conclude from the local concentrations as determined by sequencing methods, on the real total amount of modified residues, specifically in species with complex genome structure, high ploidies, and lots of repetitive sequences. That is why one needs to measure these total concentrations with a direct and quantitative analytical method, as we have done it and our paper, and not extrapolate these values from indirect measures, like a genome sequence.

– The method used to calculate DNA modification percentage, represented as normalized over G and T, does not take into account that the relative amount of G and T can differ in the respective genome/organism analyzed. When making comparisons directly across different species, the authors should consider to normalize and scale the total number of G or T present in each sample.

We apologize for any confusion and if there was lack of clarity. In this method, we have of course not only measured the modified bases but also the unmodified ones (C, T, G and A). Therefore, the variation in the amount of G or T is reflected in the measurements and shouldn't influence the DNA modification percentage, and we can accurately represent the relative levels of modifications, irrespective of genome size.

– The authors claim for Figure 3B "Our dataset shows that 5hmdC is detected in a broad range of vertebrate tissues except for spleen, but reaches significantly higher concentrations specifically in samples from the CNS". Even if the trend is clear the authors should consider using some statistical test to claim significant differences.

Assuming a normal distribution, and have applied a simple T-test, which showed that the difference is highly significant.

– In Figure 4B the authors state "At the phylum level the patterns were more prominent in Proteobacteria, containing more N6mdA than 5mdC, while a reverse trend of more 5mdC than N6mdA was observed for Bacteroidetes and Firmicutes". The reverse trend of Bacteroidetes and Firmicutes is not present as the median of the box plot is lower for 5mdC compare to N6mdA and the data have just bigger dispersion in that Phylum. As already mentioned, the authors should perform some statistical test to claim differences.

Assuming a normal distribution, and have done analysis of variance, which showed that the difference is highly significant. The p values are given in the figure description.

Minor comments:– For some of the plant samples, the authors use seeds as they represent an easier and cleaner source of genomic DNA. Is the analysis of DNA modifications in seeds comparable to "adult" fully developed tissues? Or is it different, as it happens when comparing gametes to adult tissues in animals?

We would like to correct the reviewer that we use plant seedlings (not seeds). This unlike gametes have differentiated tissues. Our (admittedly limited) data of different plant tissues suggests that the total overall concentrations is quite similar between different tissues, a situation that is also observed in mammals. We discuss this in the revision.

– Figure 4.B-C is not described in a linear way in the text, not reflecting the order in which the panels are positioned. Authors should consider reshuffling either the text or panels.

We apologize for the lack of clarity. We have performed the necessary corrections.

– In the text, there are many references to "Methods" section but this section is missing in the main text and that information are kind of reported altogether within the supplementary file. The authors should consider creating an appropriate Material and Method section in the main text.

As the material section is quite large due to large sample size with their respective protocols, we had to move this section to the supplementary information file. But to aid the reader find the sections easier, we have now included the important aspects of the section in the main manuscript.

– The common names and species names are used interchangeably; for readers to keep track, it would help to have a table listing the common names with the species names.

We apologize for the inconsistency in the usage. To simplify, we have reverted to the usage of only scientific binomial names of all the species used in this study.

– The authors should consider investigating deeper some of the interesting findings they observed in the group of bacteria hosted by higher eukaryotes that showed higher 5mdC compared to the free-living corresponding bacteria. This could be of interest not only at the epigenetic or molecular level but could also harbor a clinical relevance since some of them are able to induce human pathologies.

As this manuscript was intended only as a resource manuscript and microbiology not being the specialty of our laboratory the request of the reviewer is out of scope- we hope however, that exactly such investigations are stimulated by our resource.